# WP1234—A Novel Anticancer Agent with Bifunctional Activity in a Glioblastoma Model

**DOI:** 10.3390/biomedicines10112799

**Published:** 2022-11-03

**Authors:** Beata Pająk, Ewelina Siwiak-Niedbalska, Anna Jaśkiewicz, Maja Sołtyka, Tomasz Domoradzki

**Affiliations:** Independent Laboratory of Molecular Biology and Genetics, Kaczkowski Military Institute of Hygiene and Epidemiology, Kozielska 4, 01-163 Warsaw, Poland

**Keywords:** glioblastoma multiforme, glycolysis, 2-deoxy-D-glucose, histone deacetylases inhibitors, drug candidates, bifunctional activity

## Abstract

Glioblastoma multiforme (GBM) is the most common primary malignant brain tumor in adults with a poor prognosis. Despite significant progress in drug development, the blood–brain barrier (BBB) continues to limit the use of novel chemotherapeutics. Thus, our attention has been focused on the design, synthesis, and testing of small-molecule anticancer agents that are able to penetrate the BBB. One such compound is the D-glucose analog, 2-deoxy-D-glucose (2-DG), which inhibits glycolysis and induces GBM cell death. 2-DG has already been tested in clinical trials but was not approved as a drug, in part due to inadequate pharmacokinetics. To improve the pharmacokinetic properties of 2-DG, a series of novel derivatives was synthesized. Herein, we report the biological effects of WP1234, a 2-ethylbutyric acid 3,6-diester of 2-DG that can potentially release 2-ethylbutyrate and 2-DG inside the cells when metabolized. Using biochemical assays and examining cell viability, proliferation, protein synthesis, and apoptosis induction, we assessed the cytotoxic potential of WP1234. WP1234 significantly reduced the viability of GBM cells in a dose- and time-dependent manner. The lactate and ATP synthesis assays confirmed the inhibition of glycolysis elicited by released 2-DG. Furthermore, an evaluation of histone deacetylases (HDAC) activity revealed that the 2-ethylbutyrate action resulted in HDAC inhibition. Overall, these results demonstrated that WP1234 is a bifunctional molecule with promising anticancer potential. Further experiments in animal models and toxicology studies are needed to evaluate the efficacy and safety of this new 2-DG derivative.

## 1. Introduction

Glioblastoma multiforme (GBM) is one of the most aggressive tumor types. The five-year survival rate for GBM patients is only 6.8%, and the average length of survival for GBM patients is estimated to be only eight months [1]. Since then, no significant improvements to the first-line standard of care (SOC) therapy have been made. Moreover, almost 50% of GBM tumors are characterized by the presence of genetic mutations that reduce or inhibit the cytotoxic activity of TMZ [2]. Moreover, GBM recurrence is almost inevitable, and the prognosis remains poor, with a median survival of up to 15 months. In the case of recurrent GBM, treatment options are limited with modest effectiveness [3]. Significant hopes were raised when, in 2009, the FDA granted accelerated approval to bevacizumab injection (Avastin) as a single agent for patients with progressive GBM following SOC treatment [4]. Unfortunately, it was confirmed that bevacizumab treatment does not prolong the overall survival of GBM patients [5]. Thus, GBM remains one of the most unmet medical needs. Developing new anti-GBM drugs is challenging due to the limited penetration of molecules through the blood–brain barrier (BBB) and significant tumor heterogeneity with various sensitivity to cytotoxic agents. As reported recently, significant efforts have been made to develop new therapies targeting primary and recurrent GBM [6]. 

The presence of the BBB limits the possibility of systemically administering more specific and efficient anticancer drugs, such as recombinant proteins, cytotoxins, monoclonal antibodies, and others. This property results from the presence of epithelial-like tight junctions within the brain capillary endothelium. Small molecules with a molecular weight below 400 Da and less than eight hydrogen bonds are able to cross the BBB using a lipid-mediated free diffusion process [7]. Unfortunately, the majority of chemical drugs lack these properties.

Our attention has been focused on small molecules that are able to penetrate the BBB. 2-DG is an analog of glucose and is transported across the BBB by the glucose carrier system, mainly glucose transporter 1 (GLUT1) [8]. 2-Deoxy-D-glucose (2-DG) is a well-known glycolysis inhibitor with confirmed cytotoxic action in various cancer models, including GBM [9]. GBM cells are susceptible to 2-DG action, since glycolysis is the main pathway of glucose utilization and ATP synthesis [10]. Despite numerous preclinical and clinical studies showing promising 2-DG anticancer effects, its poor pharmacokinetic properties (rapid metabolism, short half-life, and competition with glucose for glucose transporters) limited its clinical approval [11]. Several chemical modifications have been made to overcome the above-mentioned deficiencies, and new 2-DG derivatives have been synthesized. WP1122 (3,6-di-O-acetyl-2-deoxy-D-glucose) appeared as the most promising 2-DG derivative. In contrast to 2-DG, its prodrug WP1122 is transported into the cells and through the BBB via passive diffusion, not GLUTs. After entering the cells, WP1122 is metabolized via intracellular esterases releasing 2-DG. 2-DG is further phosphorylated at the C-6 hydroxyl group and trapped inside the cell [12]. Notably, the half-life of 2-DG generated from WP1122 metabolism is nearly twice as much as for 2-DG alone [12]. Moreover, its peak plasma concentration also significantly increases (almost three times). Further, in animal studies, WP1122 provided a much higher tissue- and organ-retention of 2-DG compared to 2-DG that was administrated directly [12].

Regardless of upregulated glycolysis, GBM cells are characterized by significant epigenetic alterations. It was shown that histone acetylation abnormalities are associated with cancer development and progression, including GBM [13]. Histone acetylation leads to chromatin relaxation and transcription process activation, whereas deacetylation causes chromatin condensation and the creation of heterochromatin with repressed gene transcription [14]. It was previously proved that in tumorigenesis, the whole proteome acetylation level is significantly impaired due to the upregulated level of histone deacetylases (HDACs) [15]. In general, the inhibition of the HDAC activity represses tumor growth and induces cancer cell death, whereas it does not affect normal tissue [14]. Therefore, HDAC inhibitors (HDACis) are also considered as anti-GBM agents. One of the most promising HDACis, able to cross the BBB, is Vorinostat, which is currently being evaluated in several clinical trials [6]. Other HDACis that could translocate the BBB and reach GBM cells are sodium butyrate (NaBt) and valproic acid (VPA) [16,17]. Recently, we showed that inhibition of the glycolysis process (using 2-DG or WP1122) or HDAC activity (using NaBt or sodium valproate (NaVPA)) strongly induced GBM cells’ death [18]. Furthermore, the concomitant treatment with 2-DG/WP1122 + NaBt/NaVPA synergistically eliminated GBM cells [18]. 

Combination therapy with two or more therapeutic agents is a cornerstone of current anticancer therapy. The amalgamation of drugs enhances efficacy compared to the monotherapy [19]. However, the clinical implementation of a new combined treatment with two or more novel compounds is time-consuming and costly. According to current FDA regulations, a drug candidate’s safety must first be examined in the monotherapy regimen. Additionally, especially challenging appears to be in vivo delivery to the cancer cells of two agents at the required time and concentration. Thus, Prof. Priebe’s team proposed a new solution using a bifunctional prodrug compound that, when metabolized inside the cells, releases active molecules able to modulate more than one signaling pathway.

WP1234 (2-deoxy-3,6-di-O-(2-ethyl)butyryl-D-glucose) (Figure 1) is a novel 2-DG ester with two ethyl butyrate groups, which was designed and synthesized in Prof. Waldemar Priebe’s laboratory. 

2-Ethylbutyric acid is structurally close to VPA, an anti-epileptic drug that is widely used to treat or prevent the perioperative seizures associated with GBM [20]. Among the postulated mechanisms of VPA action, the modulation of HDAC activity was confirmed [21]. As shown by Tsai et al. [20], VPA potentiated temozolomide efficacy via downregulation of the O6-methylguanine-DNA methyltransferase (MGMT) enzyme expression, which plays an important role in cellular resistance to alkylating agents. It was also a potent cytotoxic agent in various cancer models [22].

It is hypothesized that when metabolized intracellularly, WP1234 releases active 2-ethylbutyric acid capable of inhibiting HDAC activity. It also releases 2-DG, which acts as a glycolysis inhibitor. Herein, we present the first results showing the biological anticancer effects of WP1234 mediated via its bifunctional activity in U-87 and U-251 GBM cell lines.

## 2. Materials and Methods

### 2.1. Reagents

Reagents such as cycloheximide (CHX), chloroquine (CQ), bovine serum albumin (BSA), dimethyl sulfoxide (DMSO), phenazine methosulfate (PMS), trichloroacetic acid (TCA), sulforhodamine B (SRB), acetic acid, Lactate Assay Kit, and 5-bromodeoxyuridine (BrdU) Cell Proliferation Assay Kit were obtained from Sigma-Aldrich (Saint Louis, MO, USA). 3,6-di-O(2-ethyl)butyryl-2-deoxy-D-glucose (WP1234) was synthesized at MD Anderson Cancer Institute by Professor Waldemar Priebe’s team (Houston, TX, USA). Muse™ Annexin V & Dead Cell Kit was purchased from Calbiochem, Merck Millipore (Darmstadt, Germany). HDAC Activity Colorimetric Assay Kit was obtained from BioVision (Milpitas, CA, USA). 3-(4,5-dimethylthiazol-2-yl)-5-(3-carboxymethoxyphenyl)-2-(4-sulfophenyl)-2H tetrazolium, inner salt (MTS), was purchased from Promega (Madison, WI, USA). This study’s other reagents and solvents were of the highest analytical reagent grade.

### 2.2. Cell Culture

The human glioblastoma U-87 and U-251 cells were obtained from the European Collection of Authenticated Cell Cultures (ECACC, Salisbury, Wiltshire, UK). The cells were cultured in growth medium (GM) constituted by Dulbecco’s Modified Eagle’s Medium (DMEM, Gibco-Life Technologies, Grand Island, NY, USA), which was supplemented with low (1 g/L for U-87) or high (4.5 g/L for U-251) glucose concentration, 10% (*v/v*) fetal bovine serum (FBS, Biowest, Riverside, MO, USA), Gentamicin (Sigma-Aldrich, Saint Saint Louis, MO, USA; 20 μg/mL), and Fungizone/Amphotericin B (Thermo Fisher Scientific, Waltham, MA, USA; 1 μg/mL). The cells were grown until reaching 70–80% confluence.

WP1234 was tested at various concentrations from 10 to 1000 μM for 48 or 72 h, and the IC_50_ concentration was calculated. A protein synthesis inhibitor (cycloheximide (CHX)) was used as a positive control of apoptosis induction. Due to the CHX-induced cell detachment, supernatants from CHX-treated cells were also collected. 

For viability, proliferation, and protein synthesis assays, the cells were seeded in 96-well plates at 1000 cells/well. Chloroquine (CQ), a well-known autophagosome-lysosome fusion inhibitor, was used to assess autophagy induction. 

To assess the influence of hypoxia on the GBM cells’ sensitivity to the WP1234 treatment, hypoxia-mimicking conditions were applied. Four hours before the beginning of the experiment, the prolyl hydrolase inhibitor (DMOG) (50 μM) and rhodamine (Rho) (0.25 μM) were added. DMOG and Rho were present in the medium until the end of the incubation with tested compounds (48 and 72 h). Confirmation of the transcriptional activity of the HIF-1a pathway and its downstream protein expression, such as lactate dehydrogenase A (LDHA) and pyruvate dehydrogenase kinase (PDK1), have been previously described [18]. All experiments were performed at least three times with similar results.

### 2.3. Cell Viability Assessment

Cell viability was examined by evaluating the ability of the cells to convert soluble MTS solution (3-(4,5-dimethylthiazol-2-yl)-5-(3-carboxymethoxyphenyl)-2-(4-sulfophenyl)-2H-tetrazolium, inner salt) into an insoluble purple formazan (Promega, Madison, WI, USA). Briefly, after the end of the experiment with the tested compounds, the cells were incubated for 1 h at 37 °C with MTS solution (20 μL per well). After incubation, formazan color was determined using a Synergy H1 multi-plate reader at 590 nm (BioTek, Winooski, VT, USA).

### 2.4. Cell Proliferation Assessment

To assess cell proliferation, U-87 and U-251 cells were seeded in a 96-well plate at 1000 cells/well. BrdU Cell Proliferation Assay Kit determined DNA synthesis according to the manufacturer’s protocol (Sigma-Aldrich, Saint Louis, MO, USA). Briefly, 1× BrdU solution (20 μL) was added to the cells for 24 h. Next, the cells were washed with phosphate-buffered saline (PBS), fixed (30 min), and further incubated with mouse anti-BrdU-antibody (1.5 h, room temperature (RT)). Afterward, the cells were washed with PBS and incubated with anti-mouse horseradish peroxidase (HRP) IgG (1:2000) (0.5 h, RT). HRP substrate (tetramethylbenzidine, TMB) was added for 30 min at RT to initiate the detection. The reaction was stopped with a Cell Proliferation Assay Stop Solution. The absorbance was determined with a Synergy H1 microplate reader (450 nm) (BioTek, Winooski, VT, USA).

### 2.5. Protein Synthesis Assessment

Cellular protein content was measured with the sulforhodamine B (SRB) assay. The applied protocol has been optimized for the cytotoxicity screening of adherent cells in a 96-well format, as described by Orellana and Kasinski [23]. 

After the end of the treatment with the tested compounds, cell monolayers were fixed with 10% (*w*/*v*) trichloroacetic acid and stained for 30 min. Next, the excess dye was removed by gentle washing with 1% (*v*/*v*) acetic acid. The protein-bound SRB was dissolved in 10 mM of Tris base solution (pH 10.5). Color determination was measured using a Synergy H1 multi-plate reader (510 nm) (BioTek, Winooski, VT, USA).

### 2.6. Cell Apoptosis Assessment

Apoptosis induction was assessed using the Muse™ Annexin V and Dead Cell Kit (Merck Millipore, Guyancourt, France) according to the manufacturer’s protocol. The cells were treated with various concentrations of WP1234. CHX (20 μM), a protein-synthesis inhibitor, was used as a positive control. Next, the cells were trypsinized (GibcoTM—Life Technologies, Grand Island, NY, USA), centrifuged (3000 rpm, 5 min, RT), and washed with PBS. Next, the cells were resuspended in PBS and mixed with the Muse™ Annexin V and Dead Cell reagent. After 20 min (RT, in the dark) of incubation, the assay results were measured using the Muse™ Cell Analyzer (Luminex Corporation, Austin, TX, USA). 

### 2.7. Transmission Electron Microscopy (TEM)

After the end of the experiment with the tested compounds, the cells were fixed in 2% paraformaldehyde (PFA) with 2.5% glutaraldehyde in 0.1 M of sodium cacodylate buffer (pH 7.4) (1 h, 4 °C). Next, the cells were washed with the 0.1 M sodium cacodylate buffer and post-fixed with 1% OsO4/0.1 M of sodium cacodylate buffer (1 h, RT). Afterward, the cells were dehydrated in a graded ethanol alcohol series and for 10 min in propylene oxide. Next, the cells were embedded in Epon 812. Ultrathin sections were placed on copper grids, dried, and stained with 4.7% uranyl acetate (10 min) and lead citrate (2 min). The grids were investigated with a JEOL JEM 1400 (JEOL Co., Tokyo, Japan) electron microscope. TEM analyses were performed in the Laboratory of Electron Microscopy, Nencki Institute of Experimental Biology of the Polish Academy of Sciences, Warsaw, Poland.

### 2.8. Whole-Cell Lysates Preparation

The cells were grown in 100 mm-diameter culture Petri dishes. To obtain whole-cell lysates, 1 mL of ice-cold PBS was added, and the cells were immediately scraped and centrifuged (×10,000 *g* for 10 min, 4 °C). The obtained cell pellet was lysed with 1.0 mL of radioimmunoprecipitation assay (RIPA) buffer (1× PBS, 10 mL/L of Igepal CA-630, 5 g/L of sodium deoxycholate, 1 g/L of SDS, supplemented with 0.4 mM of phenylmethylsulfonyl fluoride (PMSF), 10 μg/mL of aprotinin, and 10 μg/mL of sodium orthovanadate (Sigma-Aldrich, St. Louis, MO, USA)). The cells were further broken up by repetitive triturating with a syringe with an attached needle (0.6 mm diameter). The cell suspensions were incubated on ice (4 °C) for 30 min and afterward centrifuged for another 5 min (4 °C, ×10,000 *g*). The viscous solution was divided into smaller volumes and stored at −80 °C until use.

### 2.9. Lactate Synthesis Assessment

Colorimetric Lactate Assay Kit (Sigma-Aldrich, Saint Louis, MO, USA) was used to assess the intracellular and extracellular levels of lactate synthesized during the glycolysis process. The cells were grown on 60 mm-diameter culture Petri dishes. After the WP1234 treatment, the cells were washed with ice-cold PBS, scraped off, and centrifuged (4 °C, ×10,000 *g*, 6 min). The obtained cell pellets were resuspended in ice-cold PBS (500 μL) and centrifuged once again (4 °C, ×10,000 *g*, 6 min). The intracellular lactate dehydrogenase was removed using Pierce Protein Concentrators with polyethersulfone (PES) (Thermo Fisher Scientific Invitrogen, Waltham, MA, USA). The obtained soluble fractions were used for intracellular lactate quantification. To evaluate the extracellular lactate level, 50 μL of cell culture medium was diluted and transferred to a 96-well plate in triplicates. Next, 50 μL of Master Reaction Mix (46 μL of Lactate Assay Buffer + 2 μL of Lactate Enzyme Mix + 2 μL of Lactate Probe) was added, and the plates were incubated in the dark (15 min). Absorbance was measured using a Synergy H1 multi-plate reader (570 nm) (BioTek, Winooski, VT, USA).

### 2.10. Assessment of ATP Synthesis 

ATP synthesis was assessed with a commercial colorimetric/fluorometric assay (Sigma-Aldrich, Saint Louis, MO, USA). Briefly, after the end of the experiment, whole-cell protein lysates were prepared using 100 μL of ATP Assay Buffer, transferred into protein concentrators (Pierce Protein Concentrators PES, 10 K, MWCO) (Thermo Fisher Scientific, Waltham, MA, USA), and centrifuged (4 °C, ×12,000 *g*, 10 min). Next, 50 μL of supernatant was transferred to 96-well plates in triplicate, and 50 μL of Reaction Mix (44 μL of ATP Assay Buffer + 2 μL of ATP Probe + 2 μL of ATP Converter + 2 μL of Developer Mix) was added. The plates were further mixed for 3 min and then incubated for 30 min in the dark. Color determination was measured at 570 nm using a Synergy H1 multi-plate reader (BioTek, Winooski, VT, USA). 

### 2.11. HDAC Activity Assay

HDAC activity was examined with a colorimetric assay (BioVision, Milpitas, CA, USA) according to the manufacturer’s protocol. Briefly, after the end of the experiment, whole-cell protein lysates were prepared (Section 2.8), and protein concentration was assessed. A total of 100 μg of protein lysates was incubated (90 min, 37 °C) with 10 μL of HDAC Assay Buffer and 5 μL of HDAC colorimetric substrate. The reaction was stopped with 10 µL of Lysine Developer. After the following samples’ incubation (30 min, 37 °C), the color determination was measured with a Synergy H1 multi-plate reader (400 nm) (BioTek, Winooski, USA). As a positive control, cell lysates from the untreated cells were used with 2 μL of Trichostatin A (TSA), a potent HDAC inhibitor.

### 2.12. Western Blot Analysis

For the Western blot analysis, protein levels in the whole-cell lysates were adjusted to a 2 μg/μL concentration. Next, Laemmli sample buffer (4x concentrate, Bio-Rad, Hercules, CA, USA) was added, and the samples were denatured (5 min at 95 °C). The samples were separated on 15% polyacrylamide gels and transferred to 0.2 μm polyvinylidene difluoride membranes (PVDF, Bio-Rad, Hercules, CA, USA). Next, the membranes were blocked in 1% (*w*/*v*) bovine serum albumin (BSA) (Sigma-Aldrich, Saint Louis, MO, USA) solution in Tris-buffered saline containing 0.1% (*v*/*v*) Tween 20 (TBS-T). The membranes were further incubated with the primary antibodies: mouse anti-actin (1:2500) or rabbit anti-microtubule-associated protein 1B/2B light chains 3B (MAP-LC3) (1:1000) (Cell Signaling Technology, Danvers, MA, USA), and subsequently, with the HRP-linked secondary antibodies (anti-mouse or anti-rabbit IgG (1:1000) (Cell Signaling Technology, Danvers, MA, USA)). The reaction was developed using an ECL kit (WesternBright Sirius Chemiluminescent Detection Kit, Advansta Inc., USA). Bands detection and densitometric quantification were performed using the G-Box system (Syngene, Cambridge, UK). 

### 2.13. Statistical Analysis

The results were analyzed using GraphPad PrismTM version 8.0 software, with a one-way ANOVA test with Tukey’s multiple comparisons and/or two-way ANOVA test, followed by Bonferroni’s multiple comparisons (GraphPad Software Inc., San Diego, CA, USA). Error bars indicate the standard error of the mean (S.E.M.). Significant differences between the means are indicated as follows: * *p* < 0.05, ** *p* < 0.01, *** *p* < 0.001. The results represent the means of at least three independent experiments.

## 3. Results

### 3.1. WP1234 Exerts a Significant Cytotoxic Effect on GBM Cells

MTS viability (Figure 2), BrdU proliferation (Figure 3), and SRB protein synthesis (Figure 4) assays showed that WP1234 treatment exerted a cytotoxic effect on U-87 and U-251 GBM cells in a dose- and time-dependent manner. CHX (20 μM), a protein synthesis inhibitor, was used as a positive control in all these assays. Additionally, IC_50_ values were calculated for U-87 cells (750 μM (48 h); 500 μM (72 h)) and for U-251 cells (500 μM (48 h); 250 μM (72 h)).

It is postulated that the limited oxygen concentration and hypoxia make GBM cells almost utterly dependent on glycolysis-mediated glucose utilization. Thus, we assumed that in hypoxia-like conditions, the cytotoxic action of WP1234, which is a 2-DG derivative that can also inhibit glycolysis, should be more enhanced compared to under normoxia conditions. To examine the influence of hypoxia on WP1234 action, hypoxia-like conditions were induced using non-toxic concentrations of DMOG (50 μM) and rhodamine (Rho) (0.25 μM) (for more details, please refer to our previous paper [18]).

According to the MTS viability assay, selected WP1234 treatments (100 and 500 μM) significantly reduced U-87 cell viability compared to normoxic conditions (Figure 5, left panel). However, it should be underlined that the observed differences in U-87 cell viability did not exceed 10%. On the other hand, in U-251 cells, WP1234 cytotoxicity in hypoxia-like conditions was not significantly different from normoxia treatment (Figure 5, right panel). The obtained results indicate that U-87 and U-251 GBM cells characterize the highly glycolytic phenotype even in normoxia conditions. Our observations are in agreement with the well-known fact that GBM, along with pancreatic cancer, represents the most glycolytic tumors [9]. 

### 3.2. WP1234 Inhibits Lactate Production

To verify the efficiency of glycolysis inhibition, the production of intracellular and extracellular lactate was examined using Lactate Assay Kit (Sigma-Aldrich, Saint Louis, MO, USA), after 72 h of WP1234 (100–500 μM) treatment. We found that WP1234 downregulated the levels of lactate in U-87- and U-251-obtained culture medium in a dose-dependent manner in comparison to untreated cells (Figure 6). In the U-87 and U-251 cell lysates, the inhibition of lactate synthesis was rather constant and was estimated at about 50%. The observed results indicate the reduction in glucose utilization in the glycolysis pathway in response to WP1234 treatment. 

### 3.3. WP1234 Inhibits ATP Synthesis 

To confirm that WP1234 inhibited the glycolysis process, the ATP synthesis in cell lysates was examined using ATP Assay Kit (Sigma-Aldrich, Saint Louis, MO, USA) after 72 h of WP1234 (100–1000 μM) treatment. As shown in Figure 7, WP1234 dose-dependently diminished ATP synthesis in U-251 cells. An interesting observation was the significant increase in ATP levels after treatment with the lower doses of WP1234 in U-87 cells. However, that response needs to be further verified. Higher doses of WP1234 (500–1000 μM) significantly reduced ATP synthesis also in U-87 cells. The obtained results correspond with the lactate synthesis assay observations and confirmed the dose-dependent inhibition of glycolysis.

### 3.4. WP1234 Cytotoxicity Is Mediated via Apoptosis Induction 

To verify the involvement of apoptotic cell death in GBM cells exposed to WP1234, Annexin V Cell Death Kit was used. CHX (20 μM) was utilized as a positive control. As shown in Figure 8, the observed reduction in viability, proliferation, and protein synthesis corresponds with apoptosis induction in a dose-dependent manner. 

### 3.5. WP1234 Modulates HDAC Activity 

To verify the ability of WP1234 to modulate HDAC enzymatic activity, HDAC Activity Colorimetric Assay Kit (BioVision) was utilized. As shown in Figure 9, WP1234 (100–1000 μM) inhibited HDAC activity in a dose-dependent manner, which could lead to changes in gene transcription. Trichostatin A (TSA), a potent HDACi, was used as a positive control.

### 3.6. Autophagy Is Not Engaged in WP1234-Mediated Cytotoxic Action 

Autophagy plays dual roles in tumor growth and contributes to tumor cells proliferation or suppression, depending on the cancer stage and type. We hypothesized that WP1234 metabolite—2-DG acts as a glycolysis inhibitor, and thus, could activate the autophagy process as an alternative energy production pathway. To examine the involvement of autophagy in WP1234 cytotoxic action, chloroquine (CQ) (non-toxic concentration of 10 μM) was used (Figure 10). CQ affects the acidic environment of the lysosome and inhibits the binding of autophagosomes to lysosomes. Consequently, the accumulation of a large number of degraded proteins, including MAP-LC3 (autophagosome marker) in cells, is induced [24]. The inhibition of autophagosomes and lysosomes fusion blocks further autophagy processing.

Surprisingly, we found that autophagy is not involved in WP1234 cytotoxic action. CQ addition did not significantly affect the U-87 and U-251 cell viability when treated with the WP1234 IC_50_ concentration (72 h) (Figure 10). Moreover, the Western blot analysis did not detect significant changes in MAP-LC3 protein level in response to WP1234 treatment (Figure 11). 

On the other hand, the TEM analysis revealed the presence of numerous autophagic vacuoles (AV), especially autophagolysosomes, in U-87 cells treated with WP1234 (500 μM, 12 h) treatment (Figure 12). However, AVs were not observed in the cell ultrastructure after 8 h of incubation with WP1234. After 8 h of incubation, visible changes in the mitochondria’s ultrastructure were observed, especially significant condensation with a loss of cristae structure. The obtained TEM observations suggest that 2-DG derived from WP1234 metabolism can inhibit the glycolysis process and upregulate the autophagy pathway. However, other cellular effects, such as the lack of ATP synthesis—and probably, changes in gene transcription due to HDAC activity restriction—activate other autophagy-independent apoptotic pathways. We also hypothesize that autophagy activation is an early response to glycolysis downregulation. Thus, autophagic vacuoles were observed with TEM after 12 h of incubation. Worthy of note, the viability of WP1234-treated cells was examined after 48 and 72 h of incubation. Further studies are crucial to identify the detailed molecular mechanism of WP1234’s influence on mitochondrial function in GBM cells.

## 4. Discussion

GBM treatment options are still very limited, and the overall efficacy is poor. Despite the tremendous progress in anticancer drug development, including targeted therapy, biologics, and immunotherapy, the GBM SOC has not changed for decades. Recently, we summarized the current clinical trials of GBM drug candidates [6]. Small molecules that are able to cross the BBB and reach brain tissue are the most common types of GBM drugs being pursued. Among the tested compounds are glycolysis inhibitors, such as dichloroacetate, and HDAC inhibitors, such as Vorinostat [6]. 

We recently analyzed the anticancer effects of simultaneous inhibition of glycolysis and HDAC activity in the GBM model. We found that the combined treatment of 2-DG or WP1122 with NaBt or NaVPA synergistically induced GBM cell death [18]. As far as we know, this was the first evidence showing that the simultaneous targeting of cells’ metabolism and gene transcription is an effective anti-GBM strategy. However, we are aware that combined therapy’s clinical approval is challenging and time-consuming. Based on previous results, Prof. Priebe’s group synthesized a new 2-DG derivative—WP1234, containing two 2-ethylbutyrate groups in its structure. After entering cancer cells, WP1234 potentially undergoes enzymatic cleavage with esterases, releasing active 2-DG and ethyl butyrate, thus, simultaneously modulating glycolysis and HDAC action. The schematic representation of the hypothesized WP1234 intracellular mechanism of action is presented in Figure 13. 

Our studies confirmed that WP1234 is able to inhibit the glycolysis pathway (evaluated via ATP and lactate synthesis) and HDAC activity in a time- and dose-dependent manner. The cytotoxic action of WP1234 is mediated via apoptosis induction. As far as we know, no other available data have identified another agent with such bifunctional action. 

2-DG is a well-known glycolysis inhibitor with confirmed cytotoxic potency in several cancer models [9]. Unfortunately, its clinical approval has not been finalized due to its poor drug-like properties, especially the rapid turnover and the competition with blood glucose for glucose transporters (GLUTs) [9]. On the other hand, butyrate has effectively induced senescence and inhibited the invasiveness of GBM cells [25]. It was further confirmed by Engelhardt et al. [26] in studies using an in vivo GBM rat model. Moreover, several studies showed that combined treatment with butyrate and already available drugs (such as docetaxel or cisplatin) significantly improved their therapeutic effects [27,28,29]. Our research confirmed that targeting cell metabolism and epigenetic modifications is a powerful anti-cancer strategy. We are aware that our in vitro data need to be further confirmed in animal studies, and WP1234 toxicology must also be assessed. 

Our TEM studies revealed an interesting observation. We found that in response to WP1234, especially in the early phase, significant modifications of the mitochondria ultrastructure (condensation, shrinkage, loss of cristae, and fusion) occurred. Our studies previously observed similar changes after 2-DG and WP1122 treatment, whereas NaBt and NaVPA did not affect mitochondria morphology [18]. These similar findings suggest that WP1234-mediated mitochondria abnormalities are the consequence of 2-DG release. Karbowski et al. [30] reported that mitochondria fusion is a protective mechanism in response to apoptosis induction.

Further, Shiratori et al. [31] showed that glycolysis inhibition in PANC-1 pancreatic cancer cells, HeLa cervical cancer cells, and A549 lung cancer cell lines induced changes in the mitochondria structure and induction of mitochondria fusion. Unfortunately, the authors did not verify the molecular mechanism leading to the observed mitochondria alterations. Further studies are needed to clarify whether the observed mitochondria changes are a protective versus cytotoxic mechanism in response to WP1234 action. 

In conclusion, the presented results are promising and support the strategy of chemically modifying 2-DG and synthesizing novel, more potent 2-DG prodrugs or analogs. Numerous studies have confirmed that targeting cancer cell metabolism, especially highly glycolytic types like GBM, represents an efficient strategy for cancer cell elimination. The concept of concomitantly targeting other intracellular targets (such as HDAC, but theoretically many others) opens new possibilities for multifunctional compounds. Multifunctional compounds could be more effective and target various populations of cancer cells by modulating multiple signaling pathways. Moreover, the clinical implication of such compounds could be safer for patients and limit the possible side effects or interactions between metabolites. Prof. Priebe’s team designed several 2-DG derivatives containing acetyl, ethyl-butyrate, and other chemical groups that are considered drug candidates for GBM therapy. Their biological activity is being intensively tested, and new lead compounds will be selected soon. 

## Figures and Tables

**Figure 1 biomedicines-10-02799-f001:**
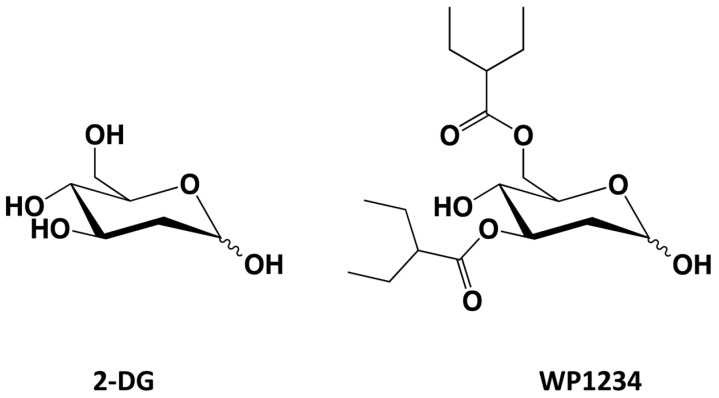
Chemical structures of 2-DG and WP1234.

**Figure 2 biomedicines-10-02799-f002:**
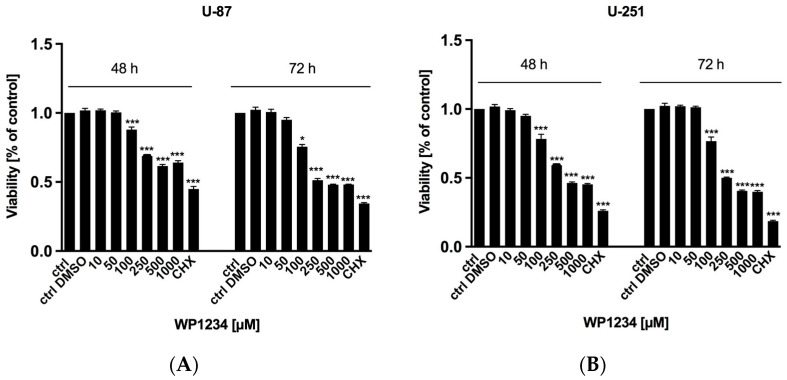
MTS assay showing the viability of (**A**) U-87 and (**B**) U-251 cells after 48 and 72 h treatment with various concentrations of WP1234 (10–1000 μM). As a reference cytotoxic agent, CHX (20 μM) was used. Significant differences between the treatment and control (untreated cells) means are indicated by * *p* < 0.05, ** *p* < 0.01, *** *p* < 0.001.

**Figure 3 biomedicines-10-02799-f003:**
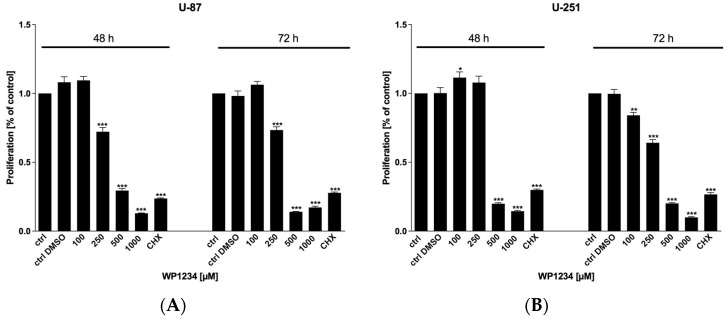
BrdU analysis of (**A**) U-87 and (**B**) U-251 cells’ proliferation after 48 and 72 h treatment with various concentrations of WP1234 (100–1000 μM). As a reference cytotoxic agent, CHX (20 μM) was used. Significant differences between the treatment and control (untreated cells) means are indicated by * *p* < 0.05, ** *p* < 0.01, *** *p* < 0.001.

**Figure 4 biomedicines-10-02799-f004:**
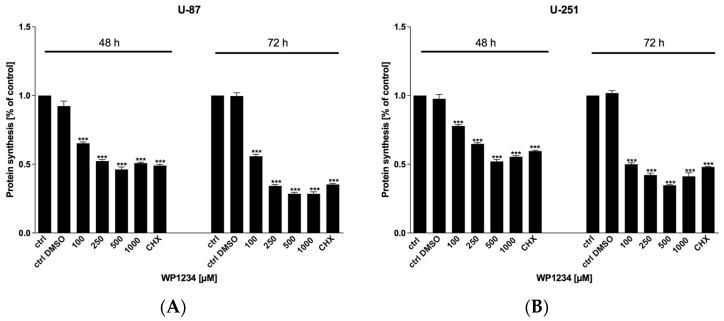
Intracellular protein synthesis of (**A**) U-87 and (**B**) U-251 in response to 48 and 72 h treatment with various concentrations of WP1234 (100–1000 μM). As a reference cytotoxic agent, CHX (20 μM) was used. Significant differences between the treatment and control (untreated cells) means are indicated by * *p* < 0.05, ** *p* < 0.01, *** *p* < 0.001.

**Figure 5 biomedicines-10-02799-f005:**
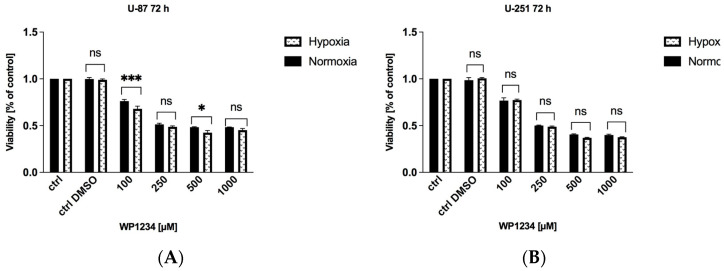
(**A**) U-87 and (**B**) U-251 cells viability after 72 h treatment with various concentrations of WP1234 (100–1000 μM) in normoxia and hypoxia-like (DMOG + Rho) conditions. Significant differences between the treatment means and control (untreated cells) value are indicated by * *p* < 0.05, ** *p* < 0.01, *** *p* < 0.001; (ns—no significance).

**Figure 6 biomedicines-10-02799-f006:**
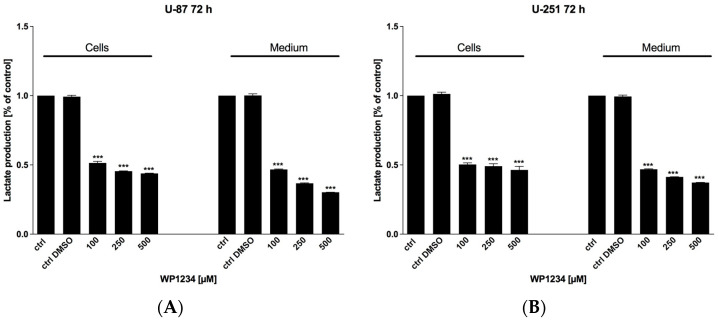
Intra- and extracellular (medium) lactate level after 72 h treatment with various concentrations of WP1234 (100–500 μM) in (**A**) U-87 and (**B**) U-251 cells. Significant differences between the treatment and control (untreated cells) means are indicated by * *p* < 0.05, ** *p* < 0.01, *** *p* < 0.001.

**Figure 7 biomedicines-10-02799-f007:**
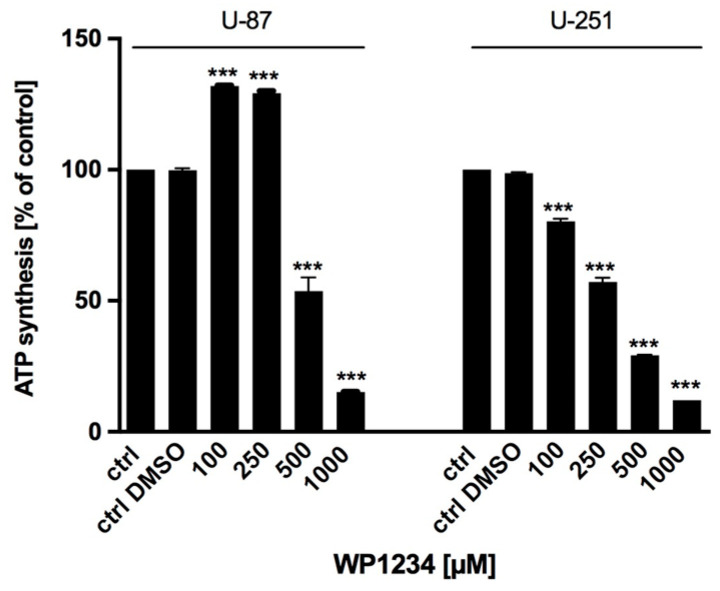
ATP synthesis assay showing glycolytic activity and ATP production of U-87 and U-251 cells after 72 h treatment with various concentrations of WP1234 (100–1000 μM). Significant differences between the treatment and control (untreated cells) means are indicated by * *p* < 0.05, ** *p* < 0.01, *** *p* < 0.001.

**Figure 8 biomedicines-10-02799-f008:**
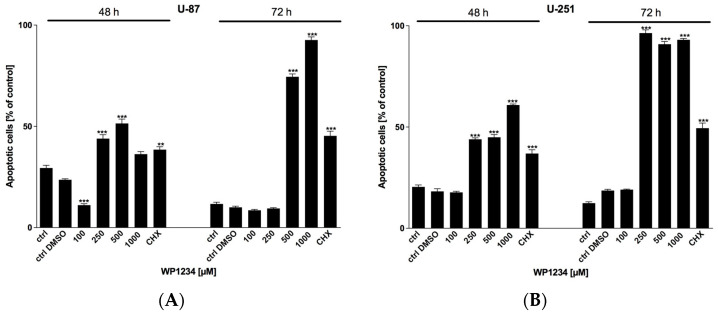
Annexin V-positive (**A**) U-87 and (**B**) U-251 cells after 48 and 72 h treatment with various concentrations of WP1234 (100–1000 μM). As a reference cytotoxic agent, CHX (20 μM) was used. Significant differences between the treatment and control (untreated cells) means are indicated by * *p* < 0.05, ** *p* < 0.01, *** *p* < 0.001.

**Figure 9 biomedicines-10-02799-f009:**
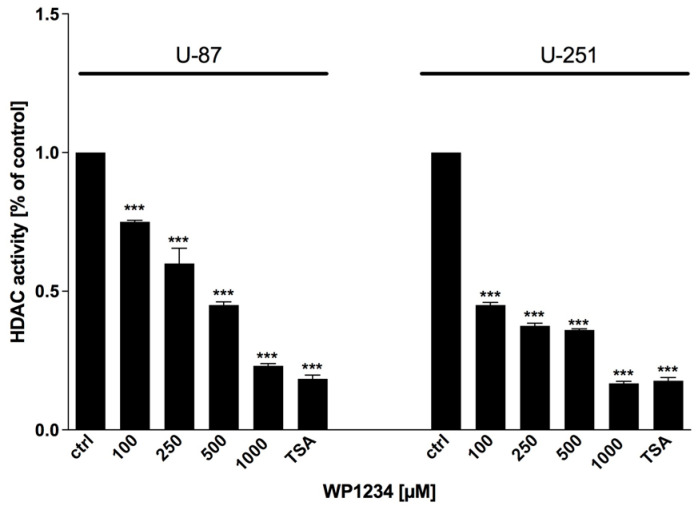
HDAC activity assay showing inhibitory effect of WP1234 (100–1000 μM) treatment after 72 h. As a positive control of HDAC inhibition, Trichostatin A (TSA) was used. Significant differences between the treatment means and control (untreated cells) value are indicated by * *p* < 0.05, ** *p* < 0.01, *** *p* < 0.001.

**Figure 10 biomedicines-10-02799-f010:**
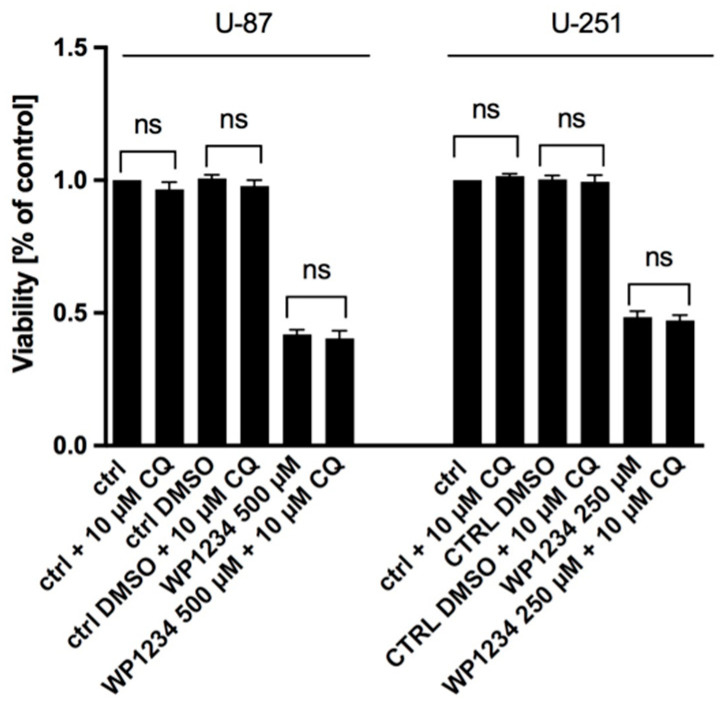
MTS assay showing viability of U-87 and U-251 cells after combined treatment of CQ (10 μM) with IC_50_ concentration of WP1234 for 72 h (ns—no significance).

**Figure 11 biomedicines-10-02799-f011:**
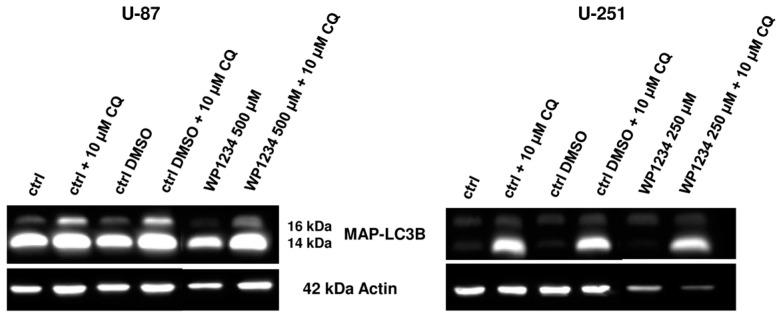
Representative Western blot analysis showing the expression of autophagy marker—MAP-LC3 protein in response to IC_50_ concentration treatment with WP1234 (500 μM for U-87 and 250 μM for U-251 cells) (72 h) alone or in combination with CQ (10 uM). Actin was used as a loading control.

**Figure 12 biomedicines-10-02799-f012:**
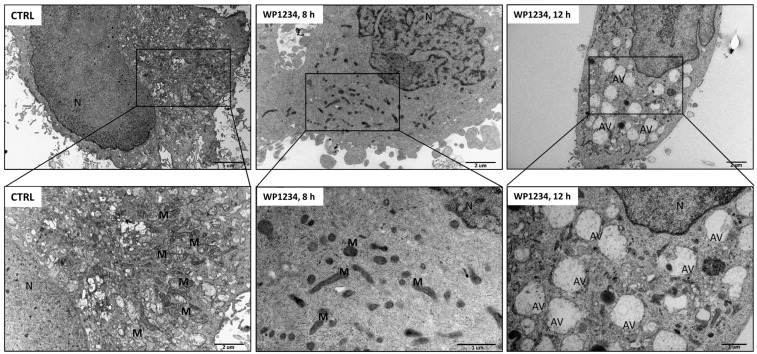
Transmission electron microscopy (TEM) images showing U-87 cells’ ultrastructure treated with IC_50_ concentration of WP1234 (500 μM; 8 and 12 h). Autophagic vacuoles are indicated as AV, mitochondria as M, and cell nuclei as N.

**Figure 13 biomedicines-10-02799-f013:**
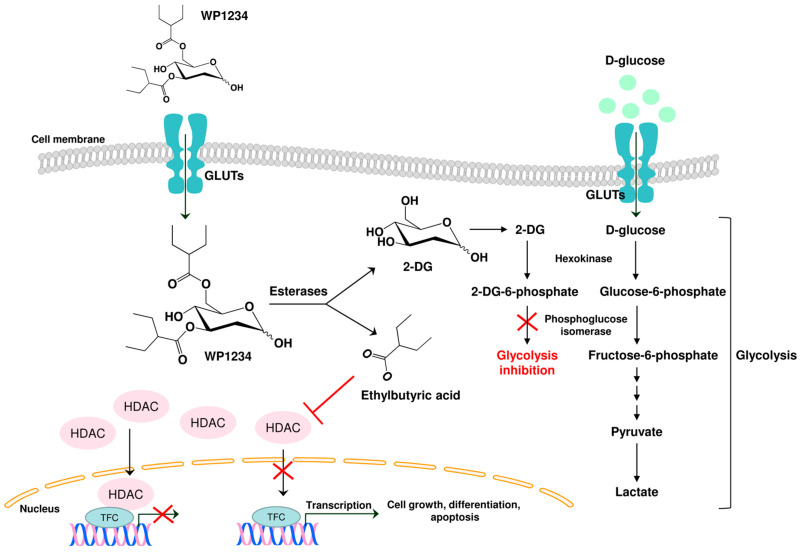
Schematic illustration of WP1234 intracellular mechanism of action. WP1234 undergoes enzymatic cleavage via esterases releasing 2-DG (2-deoxy-D-glucose) and ethylbutyric acid. 2-DG is phosphorylated via hexokinase to 2-deoxy-D-glucose-6-phosphate that could not be further converted via phosphoglucose isomerase. Consequently, the glycolysis process is inhibited. On the other hand, ethylbutyric acid acts as an HDCA inhibitor, allowing the transcription of several genes coding proteins engaged in cell growth, differentiation, and apoptosis; GLUT—glucose transporters, TCF—transcription factors complex, HDAC—histone deacetylase.

## Data Availability

The data presented in this study are available on request from the corresponding authors.

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
