# Peer review of "WP1234—A Novel Anticancer Agent with Bifunctional Activity in a Glioblastoma Model"

_biomedicines, 2022, doi:10.3390/biomedicines10112799_

Round 1
Reviewer 1 Report
biomedicines-1978504
In “WP1234 – a novel anticancer agent with bifunctional activity in a glioblastoma model”, the authors aim to characterize the activity of a prodrug – WP1234, that is metabolized into 2-DG once it enters tumor cells. 2-DG is already characterized, presenting good cytotoxicity results, however, its other characteristics (such as pharmacokinetics and competition with glucose transports) do not make it an attractive anticancer molecule.
In this paper, the authors characterize a prodrug that, besides being metabolized into 2-DG, can also inhibit HDAC activity. The conclusions are supported by results, where the authors show the activity of the compound towards GBM cells, particularly for the higher concentrations. The sections in the manuscript are well-presented, which makes it easy to follow and understand.
Major comments:
1. In the introduction is mentioned that WP1122 crosses the BBB via passive diffusion and not via glucose transporter. Is this the mechanism used by WP1234, some evidence should be presented.
2. Should include results in healthy cells, as controls, to evaluate the safety of the test compound.
3. Axis labels should have the same units, either µM or mM (0.10, 0.25, 0.50 and 1.00 mM).
4. In Figure 9 - HDAC activity the authors mention the trichostatin control, but is not presented, please include this control.
5. Would be interesting to evaluate the expression of MAP-LC3 expression to confirm autophagy.
6. References 10 and 11 are from conference abstracts, thus results in data are not available.
Minor comments by section:
Introduction
The introduction is, overall, well structured and clearly explains the challenge. The text flow is good, explaining the major concepts important to fully understand the scoop of the work. Still, I would advise giving more details on why it is difficult to translocate the BBB, giving emphasis on the importance of the molecules’ size to achieve brain penetration. The authors are focused on small molecule drugs, but lack to clearly state the advantage of small molecules over larger structures.
Additionally, I would do a quick introduction to HDAC inhibitors, as it is not clear what pathways they target.
Finally, it would be interesting to know what are the differences between the two cell lines chosen for this work, as well as the reasons behind the choice of these two.
|
Line |
Where it says |
Should be |
|
32 |
6.8 % |
6.8% (space between number and % is not necessary) |
|
35 and 36 |
“but there have not been any significant improvements to the first-line standard of care (SOC) therapy has not been significantly improved since.” |
This phrase is repetitive and should be reformulated |
|
37 |
“Almost 50% of GBM tumors characterize the presence of genetic mutations” |
Almost 50% of GBM tumors are characterized by the presence of genetic mutations |
|
41 |
“Big hopes were raised when in 2009, FDA” |
Big hopes were raised when, in 2009, FDA |
|
60 |
abovementioned |
above-mentioned |
|
63 |
“its prodrug WP1122 enters cells and the BBB” |
its prodrug WP1122 enters cells and crosses the BBB; Additionally, this phrase is very long. Consider splitting in two to facilitate reading |
|
70 |
cell |
cells |
|
72 |
“HDACi, able to cross the BBB is Vorinostat,” |
“HDACi, able to cross the BBB, is Vorinostat,” |
|
73 |
penetrate |
Translocate it’s a more appropriate term |
|
84 |
in vivo |
in vivo and change it throughout the text |
|
85 |
“cells of two agents into at the required time” |
“cells of two agents at the required time” |
|
93 |
“WP1234. .” |
WP1234. (the second dot is not necessary) |
|
101 |
“It is hypothesized that when metabolized intracellular” |
It is hypothesized that when metabolized intracellularly |
Materials and methods
Authors should add the information on cell lines origin used in this work.
|
Line |
Where it says |
Should be |
|
121 |
IC50 |
IC50 |
|
153 |
|
Specify which stop solution was used |
|
169 |
|
Missing the temperature of centrifugation |
|
194 |
used |
use |
Results
The results section is, overall, well explained and clear to the reader. The subdivision into sections allows an easier comprehension of each experiment and its objectives.
I was wondering how the authors explain the increase in proliferation, in figure 3, for the lower concentrations (U-87 cell line and 48h for U-251 cell line).
In figure 6, for U-87 cell line, in cells, the difference is not evident from the observation of the graphic.
In figure 7, it is not explained why the ATP synthesis increases with the lowest concentrations of WP1234 (for U-87 cell line).
|
Line |
Where it says |
Should be |
|
247 |
“(72 h)” |
(72 h)) – it’s missing the closure of the second parenthesis |
|
249 |
“U-87 and U-251 cells” |
U-87 and U-251 cells’ – missing possessive |
|
249 |
|
As it is mentioned in figure 6, it would be best to say here that the control are untreated cells |
|
278 |
|
Say, in the legend, that ns stands for non significant (as it is in figure 10) |
|
339 |
IC50 |
IC50 |
|
360 |
“M, cell nuclei as N” |
“M, and cell nuclei as N” |
Author Response
Thank you for the valuable comments. Please, find below our responses to Your suggestions:
Major comments:
- In the introduction is mentioned that WP1122 crosses the BBB via passive diffusion and not via glucose transporter. Is this the mechanism used by WP1234, some evidence should be presented.
- As stated in the introduction, WP1122 – diacetylated 2-DG analog can enter the cells via passive diffusion due to the presence of the acetyl groups. The presence of acetyl groups changes the properties of the chemical compound and allows it more efficient penetration through biological membranes. Good examples of such compounds are morphine and its diacetylated analog heroin. In the WP1234 structure, there are no acetyl groups thus, we do not postulate its passive diffusion. It is possible that WP1234 use glucose transporters, similarly to 2-DG, that was indicated in the Figure 12 (schematic illustration of WP1234 action). We agree with the Reviewer that it is a crucial property of the compound that should be analyzed. Thus, detailed research concerning this issue is under scrutiny.
- Should include results in healthy cells, as controls, to evaluate the safety of the test compound.
- We agree with the Reviewer’s comment and are aware that additional safety studies are crucial for further studies. As the discussion mentioned, “We are aware that our in vitro data need to be further confirmed in animal studies, and WP1234 toxicology must also be assessed”. We have not obtained results from normal cells so far, and we cannot deliver them within 10 days. Based on the promising results from the GBM model, we are continuing our research to address this important issue.
- Axis labels should have the same units, either µM or mM (0.10, 0.25, 0.50 and 1.00 mM).
- We agree with this comment. All figures have been updated according to the Reviewer’s suggestion.
- In Figure 9 - HDAC activity the authors mention the trichostatin control, but is not presented, please include this control.
- We agree with the Reviewer’s comment. The Figure 9 has been updated.
- Would be interesting to evaluate the expression of MAP-LC3 expression to confirm autophagy.
- Similarly to Figure 10 and MTS assay, we verified the expression of MAP-LC3 protein after 72 hours of treatment with IC50 WP1234 concentration -/+ CQ. Unfortunately, we did not observe an increased level of MAP-LC3 in response to WP1234. As requested, an additional figure (Figure 11) has been included in the manuscript. As mentioned in the Discussion, TEM analysis was performed after 8 h and 12 h of WP1234 treatment. It is possible that autophagy is induced as a primary response to glycolysis inhibition but has no significant impact on WP1234- mediated cytotoxicity.
- References 10 and 11 are from conference abstracts, thus results in data are not available.
- According to the MDPI article template, conference abstracts could be included in the References section. Abstract nr 10 is available online. Citation number 10 is essential for discussed issue concerning novel 2-DG analogs and was not further published as a full paper. Thus, we would like to remain it in the manuscript. However, we deleted ref number 11 because the abstract is no longer available. We agree with the Reviewer that verification of the cited data is impossible. Mentioned information was also included in the cited abstract Ref. 10. The References list has been updated. Moreover, the ref 10 notation has been corrected as suggested in the MDPI template:
Author 1, A.B.; Author 2, C.D.; Author 3, E.F. Title of Presentation. In Proceedings of the Name of the Conference, Location of Conference, Country, Date of Conference (Day Month Year).
Minor comments by section:
Introduction
The introduction is, overall, well structured and clearly explains the challenge. The text flow is good, explaining the major concepts important to fully understand the scoop of the work. Still, I would advise giving more details on why it is difficult to translocate the BBB, giving emphasis on the importance of the molecules’ size to achieve brain penetration. The authors are focused on small molecule drugs, but lack to clearly state the advantage of small molecules over larger structures.
- The importance of molecules’ size has been addressed in the Introduction, according to the Reviewer’s suggestion:
Line 50: “The presence of the BBB limits the possibility of systemic administration of more specific and efficient anticancer drugs, such as recombinant proteins, cytotoxins, monoclonal antibodies, and others. This property arises from the epithelial-like tight junctions within the brain capillary endothelium. Some small molecule compounds may cross the BBB via lipid-mediated free diffusion, especially the drug with a molecular weight <400 Da and <8 hydrogen bonds [7]. These chemical properties are lacking in the majority of small molecules drugs, and all large molecule drugs.
Our attention has been focused on small molecules that are able to penetrate the BBB. 2-DG is an analog of glucose and is transported across the BBB by the glucose carrier system, mainly glucose transporter 1 (GLUT1) [8]. 2-Deoxy-D-glucose (2-DG) is a well-known glycolysis inhibitor with confirmed cytotoxic action in various cancer models, including GBM [9]”.
Additionally, I would do a quick introduction to HDAC inhibitors, as it is not clear what pathways they target.
- A quick introduction to HDAC inhibitors has been added according to the Reviewer’s suggestion:
Line 76 “Regardless of upregulated glycolysis, GBM cells are characterized by significant epigenetic alterations. It was shown that histone acetylation abnormalities are associated with cancer development and progression, including GBM [13]. Histone acetylation leads to chromatin relaxation and transcription process activation, whereas deacetylation causes chromatin condensation and the creation of heterochromatin with repressed gene transcription [14]. It was previously proved that in tumorigenesis, the whole proteome acetylation level is significantly impaired due to the upregulated level of histone deacetylases (HDAC) [15]. In general, inhibition of the HDAC activity represses tumor growth and induces cancer cell death, whereas it does not affect normal tissue [14]. Therefore, HDAC inhibitors (HDACi) are also considered as anti-GBM agents. One of the most promising HDACi, able to cross the BBB is Vorinostat, which is currently being evaluated in several clinical trials [6]”.
Finally, it would be interesting to know what are the differences between the two cell lines chosen for this work, as well as the reasons behind the choice of these two.
- U-87 and U-251 cells are commonly used as experimental models of glioblastoma. However, these cells exhibit significant differences in their proliferation, invasion, and migration. Differentially expressed proteins between the U251 and U87 cell lines are associated with the regulation of nicotinamide nucleotide metabolism, RNA splicing, glycolysis, and purine metabolism pathways. The detailed characteristic of U087 and U-251 cells has been previously described in the following papers:
https://www.ncbi.nlm.nih.gov/pmc/articles/PMC6331579/
https://pubmed.ncbi.nlm.nih.gov/27651343/
What is important for our studies, U-251 cells represent highly glycolytic potential. It is known that glycolytic phenotype correlates with drug-resistance. We used various GBM cell lines for our studies. In this case, due to the postulated molecular mechanism of WP1234 action as a glycolysis inhibitor, we used highly glycolytic (U-251) and less glycolytic cells (U-87) to examine the antimetabolic efficacy of WP1234 in various metabolic states of GBM cells.
|
Line |
Where it says |
Should be |
Response |
|
32 |
6.8 % |
6.8% (space between number and % is not necessary) |
corrected |
|
35 and 36 |
“but there have not been any significant improvements to the first-line standard of care (SOC) therapy has not been significantly improved since.” |
This phrase is repetitive and should be reformulated |
the phrase has been corrected |
|
37 |
“Almost 50% of GBM tumors characterize the presence of genetic mutations” |
Almost 50% of GBM tumors are characterized by the presence of genetic mutations |
the phrase has been corrected |
|
41 |
“Big hopes were raised when in 2009, FDA” |
Big hopes were raised when, in 2009, FDA |
corrected |
|
60 |
abovementioned |
above-mentioned |
corrected |
|
63 |
“its prodrug WP1122 enters cells and the BBB” |
its prodrug WP1122 enters cells and crosses the BBB; Additionally, this phrase is very long. Consider splitting in two to facilitate reading |
corrected |
|
70 |
cell |
cells |
corrected |
|
72 |
“HDACi, able to cross the BBB is Vorinostat,” |
“HDACi, able to cross the BBB, is Vorinostat,” |
corrected |
|
73 |
penetrate |
Translocate it’s a more appropriate term |
corrected |
|
84 |
in vivo |
in vivo and change it throughout the text |
corrected |
|
85 |
“cells of two agents into at the required time” |
“cells of two agents at the required time” |
corrected |
|
93 |
“WP1234. .” |
WP1234. (the second dot is not necessary) |
corrected |
|
101 |
“It is hypothesized that when metabolized intracellular” |
It is hypothesized that when metabolized intracellularly |
corrected |
Materials and methods
Authors should add the information on cell lines origin used in this work.
- We agree with the Reviewer’s comment. The information about cells origin and cell culture conditions has been added:
Line 165: “The human glioblastoma U-87 and U-251 cells were obtained from the European Collection of Authenticated Cell Cultures (ECACC, Salisbury, Wiltshire, UK). Cells were cultured in growth medium (GM) constituted by Dulbecco’s Modified Eagle’s Medium (DMEM, Gibco-Life Technologies, Grand Island, NY, USA) supplemented with low (1 g/L for U-87) or high (4.5 g/L for U-251) glucose concentration, 10% (v/v) fetal bovine serum (FBS, Biowest, Riverside, MO, USA), Gentamicin (Sigma-Aldrich, Saint Saint Louis, MO, USA; 20 μg/mL), and Fungizone/Amphotericin B (Thermo Fisher Scientific, Waltham, MA, USA; 1 μg/mL). Cell were grown until 70-80% confluence”.
|
Line |
Where it says |
Should be |
Response |
|
121 |
IC50 |
IC50 |
corrected |
|
153 |
|
Specify which stop solution was used |
it should be: Cell Proliferation Assay Stop Solution; the reagent is already available in the commercial assay; Unfortunately, we do not know its composition. |
|
169 |
|
Missing the temperature of centrifugation |
it was RT, the temperature has been added. |
|
194 |
used |
use |
corrected |
Results
The results section is, overall, well explained and clear to the reader. The subdivision into sections allows an easier comprehension of each experiment and its objectives.
- Thank you for positive comment.
I was wondering how the authors explain the increase in proliferation, in figure 3, for the lower concentrations (U-87 cell line and 48h for U-251 cell line).
- The increase in the proliferation the lower concentration of WP1234 is observed in both cell lines. Although these differences were not significant and did not exceed a few %, it is possible that we observed the effect of low concentrations of DMSO that could be stimulatory. We also hypothesize that call viability reduction is primary response that consequently induces also the downregulation of more stable parameter, such as DNA content. Because the overall response in BrdU assay to WP1234 treatment corresponds with other cytotoxicity assays and apoptosis activation, we did not analyze these effects in details.
In figure 6, for U-87 cell line, in cells, the difference is not evident from the observation of the graphic.
- Thank you for this comment. We checked the statistical results once again and corrected the results description. In cells the response in both cell lines is similar and did not depend on WP1234 dose.
Line 369: “We found that WP1234 downregulated the levels of lactate in U-87 and U-251-obtained culture medium in dose-dependent manner in comparison to untreated cells (Figure 6). In the U-87 and U-251 cell lysates, the inhibition of lactate synthesis was rather constant and estimated about 50%. Observed results indicate the reduction in glucose utilization in the glycolysis pathway in response to WP1234 treatment”.
In figure 7, it is not explained why the ATP synthesis increases with the lowest concentrations of WP1234 (for U-87 cell line).
- We agree with the Reviewer’s observation and corrected the Results description to:
Line 389: “As shown in Figure 7, WP1234 dose-dependently diminished ATP synthesis in U-251 cells. An interesting observation was the significant increase in ATP levels after treatment with the lower doses of WP1234 in U-87 cells. However, that response needs to be further verified. Higher doses of WP1234 (500-1000 μM) significantly reduced ATP synthesis also in U-87 cells. The obtained results correspond with the lactate synthesis assay observations and confirmed the dose-dependent inhibition of glycolysis”.
We admit that it is difficult to find a simple answer for the observation mentioned above, especially since other assays confirmed the cytotoxic action of WP1234. We need to verify the U-87 response further. However, to clarify the presented Figure, we added such a comment in the text. It would be advantageous to analyze the ATP with another assay (if available for a multiplate reader) and compare the result. However, we are not able to make this analysis within 10 days. Overall, the higher concentration of W1234 exerted a cytotoxic effect in U-87 cells and downregulated the ATP synthesis. All analyzed assays correspond with each other showing the cytotoxic potential of WP1234 in the GBM model.
|
Line |
Where it says |
Should be |
Response |
|
247 |
“(72 h)” |
(72 h)) – it’s missing the closure of the second parenthesis |
corrected |
|
249 |
“U-87 and U-251 cells” |
U-87 and U-251 cells’ – missing possessive |
corrected |
|
249 |
|
As it is mentioned in figure 6, it would be best to say here that the control are untreated cells |
corrected |
|
278 |
|
Say, in the legend, that ns stands for non significant (as it is in figure 10) |
corrected |
|
339 |
IC50 |
IC50 |
corrected |
|
360 |
“M, cell nuclei as N” |
“M, and cell nuclei as N” |
corrected |
Reviewer 2 Report
The authors reported significant effects of the newly synthesized 2-deoxy-D-glucose analog on viability, proliferation, protein synthesis, apoptosis, lactate synthesis, and subcellular structures of U-87 and U-251 glioblastoma cells in their paper titled "WP1234 - a novel anticancer agent with bifunctional activity in a glioblastoma model." The structure and design of the experiments are appropriate.
I have only minor comments:
page 4 line 142: the authors write that 490 nm was used to detect formazan absorption. Is this correct? Normally 590 nm is used for detection of purple absorption. Please correct this.
In some results, DMSO is indicated as the control. What is referred to as control in all studies? Should it also be DMSO as vehicle?
Author Response
The authors reported significant effects of the newly synthesized 2-deoxy-D-glucose analog on viability, proliferation, protein synthesis, apoptosis, lactate synthesis, and subcellular structures of U-87 and U-251 glioblastoma cells in their paper titled "WP1234 - a novel anticancer agent with bifunctional activity in a glioblastoma model." The structure and design of the experiments are appropriate.
- Thank you for the valuable comments and appreciation of our efforts. Please, find below our responses to Your suggestions.
I have only minor comments:
page 4 line 142: the authors write that 490 nm was used to detect formazan absorption. Is this correct? Normally 590 nm is used for detection of purple absorption. Please correct this.
- We agree with the Reviewer. It should be 590 nm. The line has been corrected.
In some results, DMSO is indicated as the control. What is referred to as control in all studies? Should it also be DMSO as vehicle?
- We agree with the Reviewer. We updated the Figures and include the control DMSO, that as a vehicle for WP1234, was also tested.
Round 2
Reviewer 1 Report
biomedicines-1978504
The authors have addressed the main concerns pointed out in the previous review report and I endorse its publication.
Minor suggestion: line 56 “especially the drug with a molecular weight <400 Da and <8 hydrogen bonds”, should read “especially if the drug has a molecular weight <400 Da and <8 hydrogen bonds”